# Urinary Phthalate Biomarkers during Pregnancy, and Maternal Endocrine Parameters in Association with Anthropometric Parameters of Newborns

**DOI:** 10.3390/children9030413

**Published:** 2022-03-14

**Authors:** Henrieta Hlisníková, Branislav Kolena, Miroslava Šidlovská, Miloš Mlynček, Ida Petrovičová

**Affiliations:** 1Department of Zoology and Anthropology, Faculty of Natural Sciences and Informatics, Constantine the Philosopher University in Nitra, 94974 Nitra, Slovakia; bkolena@ukf.sk (B.K.); msidlovska@ukf.sk (M.Š.); ipetrovicova@ukf.sk (I.P.); 2Department of Nursing, Faculty of Social Sciences and Health Care, Constantine the Philosopher University in Nitra, 94974 Nitra, Slovakia; mmlyncek@ukf.sk

**Keywords:** phthalate metabolites, maternal hormones, prenatal exposure, newborns, birth outcomes, anthropometry

## Abstract

Adverse birth outcomes present risk factors resulting in neonatal morbidity and mortality. Sufficient maternal hormonal concentrations are crucial for normal foetal development. Previous studies have shown a relationship between phthalate exposure and maternal hormonal levels during pregnancy. This study aims to investigate if neonatal anthropometric parameters are associated with maternal endocrine parameters during the ≤15th week of gestation and the third trimester of pregnancy concerning phthalate exposure in pregnant women from Nitra, Slovakia. We used high-performance liquid chromatography, tandem mass spectrometry (HPLC-MS/MS), and electro-chemiluminescence immunoassay to quantify urinary concentrations of phthalates and serum concentrations of hormones and sex hormone-binding globulin (SHBG), respectively. We observed a mostly positive correlation between neonatal anthropometric parameters (gestational age, birth length, birth weight, head circumference) and maternal concentration of phthalate metabolites (*p* ≤ 0.05). The hierarchical multivariate regression results showed a statistically significant association between Apgar score at 5 min after delivery, gestational age, birth weight, head circumference, and maternal endocrine parameters during pregnancy (*p* ≤ 0.05), adjusted to phthalate metabolites. To the best of our knowledge, our study is the first to indicate that prenatal exposure to phthalates may also affect birth outcomes through interaction with the maternal endocrine system.

## 1. Introduction

Foetal growth and development are complex processes regulated by genetic and hormonal factors, nutrition, and the maternal and foetal environment [1]. The prenatal period is dependent on the maternal endocrine system [2]. More specifically, proper foetal development needs adequate levels of estradiol [3], cortisol [4], and thyroid hormones [5]. Prenatal growth and development represent sensitive windows for exposure to environmental pollutants, such as phthalates [6]. Phthalates are applied widely within the plastic industry. They are used as plasticisers to increase softness and elasticity in various plastic products. Humans are ubiquitously exposed to phthalates via skin contact by cosmetic products, breathing phthalate-contaminated air, and consumption of contaminated food and beverages (reviewed in [7]). Moreover, because of phthalates’ lipophilic properties [8], they can reach the placental barrier and expose the developing foetus [9]. Phthalates belong to endocrine disruptors. Endocrine-disrupting chemicals interact with the hormonal system of the organism, which leads to endocrine-system imbalances such as altered foetal levels of adrenal and maternal gonadal steroids or thyroid hormones (reviewed in [10]). This disruption could induce adverse development and growth during the prenatal period. Previous studies have shown both the negative and positive associations between maternal levels of phthalate metabolites and birth outcomes in newborns, such as gestational age at delivery, birth weight, birth length, and head circumference (reviewed in [11]). Based on the revealed data, it is crucial to monitor prenatal phthalate exposure to avoid maternal hormonal imbalance with a potential effect on the birth outcomes of progeny.

Although there has been research conducted to study the associations between phthalate exposure and birth outcomes or endocrine parameters, no study to our knowledge has focused on the effect of phthalate exposure mediated through maternal endocrine parameters on a newborn’s health. Therefore, our study aims to determine this relationship, since phthalates may act in the body by similar mechanisms as natural hormones [12], such as competition for the binding site on transport proteins with hormones, alternation of synthesis, metabolism of steroid and nonsteroid hormones, or alteration of a feedback loop within the hypothalamus, pituitary gland, and peripheral glands. The most studied effect of phthalates is the interaction with nuclear receptors. A possible reason is a fact that phthalates share a similar chemical structure with steroid hormones, and that is the benzene ring that binds to the receptor, acting as an agonist or antagonist (reviewed in [10,13]). These interactions may result not only in complications and worsening of the newborn’s health, but also can cause lifelong health problems.

## 2. Materials and Methods

### 2.1. Study Population

A presented study is a part of the Mother–Infant Study Cohort (PRENATAL) conducted in Nitra, Slovakia, designed to investigate the association between maternal phthalate exposure and health-related outcomes of progeny. Participation was anonymous and voluntary. Our study population consisted of 72 mother–newborn pairs from the Nitra region, Slovakia. Exclusion criteria for the potential probands were age (≤17 years old), week of pregnancy during the early stage of pregnancy (>16th week of pregnancy), risks of pregnancy, hospitalisation, severe disorder (e.g., cancer, stroke, heart attack), and living in other regions of Slovakia than Nitra. The sample collection was realised during the early stage of pregnancy (8.–15. week of gestation; n = 60) and/or during the third trimester of pregnancy (30.–38. week of gestation; n = 70) from October 2017 to December 2018 with the participation of the University Hospital in Nitra. The research was conducted following the permission of the Ethics Committee of the University Hospital in Nitra (protocol code: PRENATAL, 13 September 2017). Written informed consent was obtained from all participants. The collection and processing of blood and urine samples are described elsewhere [14].

### 2.2. Questionnaire Data Collection

A questionnaire was used to collect data about baseline characteristics of our probands, health status, and information concerning previous pregnancies. The questionnaire was also used to assess the potential sources of phthalate exposure. We asked probands about their eating, hygiene habits, and cosmetics use. For each item in the questionnaire, probands indicated whether this item had been used or not used in the last 48 h. The use or non-use of the items was scored by 1 and 0, respectively. Based on the sum of the scores of all items, we calculated the total score of consumer practices of our probands.

### 2.3. Birth Data Collection

The analysed neonatal anthropometric parameters, such as gestational age at baby delivery (gestational week), birth weight (g) and length (cm), head circumference (cm), and Apgar score, were measured by obstetricians and medical staff immediately after delivery. The Apgar score is a rapid method of examining the basic vital functions of the newborn (Table 1) immediately after birth [15]. The Apgar score is assessed by the medical staff immediately after delivery (after 1 min, 5 and 10 min). These data were provided to us through the medical record of each newborn.

Ponderal index (*CI*) was calculated based on the birth weight (g) and birth length (cm) ratio [16], according to the Equation (1):(1)$$CI=1000\times \frac{birth\;weight\;\left(g\right)}{birth\;length\;{\left(\mathrm{cm}\right)}^{3}}$$

### 2.4. Qualitative and Quantitative Analysis of Maternal Phthalate Metabolites from Urine Spots

The description of qualitative and quantitative analysis of phthalate metabolite is described elsewhere [14]. Briefly, we used high-performance liquid chromatography (HPLC) and tandem mass spectrometry (MS/MS) (Infinity 1260 and 6410 triplequad, Agilent, Santa Clara, CA, USA) to quantify the urinary concentration of 17 phthalate metabolites by the method built on the basis of previously published offline SPE and online HPLC-MS/MS methods [17,18]. The analysis was performed in Physiological Analytical Laboratory, Constantine the Philosopher University in Nitra, which has participated in the HBM4EU QA/QC programme, and its successful performance has resulted in its qualification as HBM4EU laboratory for the analysis of phthalate metabolites in human urine. Within this testing, we obtained very satisfactory Z-scores for previously meant compounds ranging from −1.1 to 0.7. The interlaboratory test conditions for a successful passing were the Z scores ≤ |2|. Detailed description of these tests is summarised in a paper written by Esteban López et al. [19]. Internal quality control was performed by analyses of 2 control materials (mixture of urine samples) with known concentrations (lower and higher concentration). The limits of quantification (LOQ) were estimated between 1 and 2.5 ng/mL. Precursor and product ions and LOQs are shown elsewhere [14].

### 2.5. Qualitative and Quantitative Analysis of Maternal Endocrine Parameters from Blood Serum Spots

Analysis of reproductive (follicle-stimulating hormone-FSH, estradiol-E2, luteinising hormone-LH, testosterone-T), stress-related (cortisol-F), and thyroid hormones (TSH-thyroid-stimulating hormone, FT3- triiodothyronine, FT4-free thyroxine), and transport protein (sex hormone-binding globulin-SHBG) was performed automatically by electro-chemiluminescence immunoassay (Elecsyssystem; Roche, Basel, Switzerland) from samples of human serum [20] in collaboration with IPBM VIVE4, Bratislava.

### 2.6. Statistics

For a description of urinary phthalate metabolite concentrations, concentrations of endocrine parameters, and birth outcomes, means with standard deviations (SDs), medians, minimum, and maximum of data were calculated. The sums of phthalate metabolites (ΣDiBP-sum of di-iso-butyl phthalate metabolites, ΣDnBP-sum of di-n-butyl phthalate metabolites, ΣDBP-sum of di-iso-butyl phthalate and di-n-butyl phthalate metabolites, ΣDEHP-sum of di(2-ethylhexyl) phthalate metabolites, ΣDiNP-sum of di-iso-nonyl phthalate metabolites, ΣLMWP-sum of low-molecular-weight phthalate metabolites, ΣHMWP-sum of high-molecular-weight phthalate metabolites) were calculated by summing the concentrations of selected phthalate metabolites, excluding metabolites whose concentrations were below the LOQ in all samples. Correlations between maternal phthalate metabolite concentrations and birth outcomes were examined by Spearman correlation analysis. According to Evans [21], the strength of the correlation is determined by the height of the correlation coefficient (r): 0.00–0.19 is very weak, 0.20–0.39 is weak, 0.40–0.59 is moderate, 0.60–0.79 is strong, and 0.80–1.0 is a very strong correlation. We used GraphPad Prism 8.0 for the visualisation of correlation matrices. The nonparametric Mann–Whitney U (Wilcoxon rank-sum) test and Kruskal–Wallis one-way analysis of variance were used to analyse differences between two or more groups in the baseline characteristics of the mother cohort associated with phthalate metabolites, endocrine parameters, and neonatal outcomes. Multivariate regression analysis was used to estimate associations between birth outcomes and maternal endocrine parameters adjusted to maternal phthalates exposure, in which phthalate metabolites and endocrine parameters were base 10 log-transformed. All statistical analyses were performed using the SPSS for Windows statistical package (version 21.0; SPSS Inc., Chicago, IL, USA) and Statistica 7 (StatSoft). A difference was statistically significant when the *p* ≤ 0.05.

## 3. Results

### 3.1. Cohort Characteristics

The cohort consisted of 72 mother–newborn pairs from the Nitra region, Slovakia. A total number of mothers involved in our study reached 70. However, two of them had twin pregnancies. In Table 2, the baseline characteristics of our mother cohort are shown.

Of our newborns, 61.11% (n = 44) were boys (average gestational age at delivery 39.14 ± 1.47 weeks; average birth weight 3421.14 ± 496.30 g; average birth length 50.14 ± 2.32 cm, average head circumference 34.74 ± 1.84 cm) and 38.89% (n = 28) were girls (average gestational age at delivery 38.75 ± 1.88 weeks; average birth weight 3196.79 ± 482.74 g; average birth length 49.11 ± 2.17 cm, average head circumference 34.04 ± 1.41 cm). Baseline characteristics of newborns’ outcomes in general and in groups of boys and girls are described in Table 3.

### 3.2. Biomonitoring of Maternal Phthalate Metabolites

We evaluated exposure to metabolites of dimethyl phthalate (DMP), diethyl phthalate (DEP), di-iso-butyl phthalate (DiBP), di-n-butyl phthalate (DnBP), di-n-pentyl phthalate (DnPeP), dicyclohexyl phthalate (DCHP), butyl benzyl phthalate (BBzP), di(2-ethylhexyl) phthalate (DEHP), di-iso-nonyl phthalate (DiNP), and di-n-octyl phthalate (DnOP) based on their metabolites analysed in the first-morning urine samples coupled with a questionnaire. We found that the urinary concentrations of detected phthalate metabolites during early stage of pregnancy in our probands were above the LOQ in 96.67% of cx-MEPP, 91.67% of samples for OH-MnBP, 91.53% for MEP, 90.00% for MiBP, 88.33% for MnBP, 83.33% for oxo-MEHP, 81.67% for OH-MEHP, 76.67% for OH-MiNP, 71.67% for OH-MiBP, 70.00% for cx-MiNP, 50.00% for MBzP, 30.51% for MMP, 25.00% for oxo-MiNP, and 0.00% for MCHP, MnPeP, MiNP, and MnOP. We found that the urinary concentrations of detected phthalate metabolites during the third trimester of pregnancy in our probands were above the LOQ in 100.00% of samples for MiBP, 98.57% for cx-MEPP and oxo-MEHP, 97.14% for OH-MiNP and OH-MnBP, 95.72% for MEP, 94.29% for OH-MEHP, 92.86% for MnBP, 85.71% for cx-MiNP, 81.43% for OH-MiBP, 50.00% for oxo-MiNP, 35.71% for MBzP, 24.29% for MMP, 5.71% for MiNP, and 0.00% for MCHP, MnPeP, and MnOP. Descriptive statistics of concentrations of phthalate metabolites (ng/mL) in our cohort are shown in Table 4.

When we compared the concentrations of phthalate metabolites during the pregnancy, we found significantly higher (*p* ≤ 0.05) concentrations of OH-MEHP, oxo-MEHP, OH-MiNP, oxo-MiNP, ∑DEHP, ∑DiNP, and ∑HMWP in the third trimester in comparison with the early pregnancy (Table 5).

### 3.3. Analysis of Maternal Endocrine Parameters

In our cohort, we evaluated the concentrations of hormones (FSH, LH, estradiol, testosterone, TSH, FT3, FT4, cortisol), and transport protein SHBG from maternal blood serum during the early stage (≤15th week of gestation) and third trimester of pregnancy. Descriptive statistics of selected maternal hormones and SHBG are shown in Table 6.

### 3.4. Analysis of Maternal Baseline Characteristics and Consumer Practices in Association with Concentrations of Phthalate Metabolites, Endocrine Parameters, and Newborns’ Outcomes

We evaluated maternal baseline characteristics as follows: maternal age, BMI, education, living area, smoking, exposure to second-hand smoke, number of previous pregnancies, using pharmaceuticals, hormonal therapy, health condition, and total score of consumer practices.

Spearman correlation analysis between scale variables and concentrations of phthalate metabolites, endocrine parameters, and newborns’ outcomes detected some significant correlations. Maternal age was negatively associated with ∑DBP (r = −0.262; *p* = 0.047) during the early pregnancy, with estradiol during the third trimester of pregnancy (r = −0.284; *p* = 0.022), and with a birth length of newborns (r = −0.236; *p* = 0.046). Maternal BMI during the early stage of pregnancy was positively associated with phthalate metabolites MnBP (r = 0.295; r = 0.024), OH-MiBP (r = 0.367; *p* = 0.005), MBzP (0.270; r = 0.040), ∑DnBP (r = 0.292; r = 0.026). In addition, maternal BMI during the early pregnancy was negatively associated with SHBG during the early stage of pregnancy (r = 0.282; r = 0.032). Maternal total score in consumer practices during the third trimester was in positive association with MEP (r = 0.298; *p* = 0.016), cx-MiNP (r = 0.248; *p* = 0.046), ∑LMWP (r = 0.282; *p* = 0.023).

The Mann–Whitney test and Kruskal–Wallis test showed significant associations between nominal and ordinal variables and concentrations of phthalate metabolites, endocrine parameters, and newborns’ outcomes. When we divided our mother cohort based on education, we noticed significantly higher cortisol levels during the third trimester of pregnancy in women with a high school education compared to women with a college/university education (*p* = 0.048). When we divided the cohort based on the living area, we noticed significantly higher levels of TSH during early pregnancy (*p* = 0.016) and the third trimester of pregnancy (*p* = 0.034) in women living in urban areas. In women using hormonal therapy, we observed significantly lower levels of MnBP (*p* = 0.045), oxo-MEHP (*p* = 0.031) during the third trimester, OH-MEHP (*p* = 0.034) during early pregnancy, and significantly higher concentrations of FT3 (*p* = 0.026) during the third trimester and cortisol (*p* = 0.047) during early pregnancy. Women using pharmaceuticals had higher levels of OH-MiBP (*p* = 0.005) and FT4 (*p* = 0.035) during early pregnancy. In women diagnosed with a thyroid disorder, we noticed significantly higher levels of MMP (*p* = 0.037) and SHBG (*p* = 0.042) during early pregnancy. In newborns of women diagnosed with a thyroid disorder, we found higher PI (*p* = 0.027) and birth length (*p* = 0.043). In women diagnosed with ovarian disorder (ovarian cyst), we observed significantly lower levels of FT4 (*p* = 0.006) and cortisol (*p* = 0.035) during the third trimester of pregnancy. We noticed significantly higher levels of MEP (*p* = 0.03), MiBP (*p* = 0.049), OH-MnBP (*p* = 0.049), ∑DiBP (*p* = 0.039), ∑DnBP (*p* = 0.046), ∑DBP (*p* = 0.035), ∑LMWP (*p* = 0.01) during the early pregnancy; MnBP (*p* = 0.016), OH-MiBP (*p* = 0.02), MBzP (*p* = 0.038), cx-MiNP (*p* = 0.007), ∑DnBP (*p* = 0.038), and ∑DBP (*p* = 0.048) during the third trimester of pregnancy in women exposed to second-hand smoke. In addition, we observed significantly higher gestational age (*p* = 0.037), birth weight (*p* = 0.014), and head circumference (*p* = 0.016) in newborns whose mothers were exposed to second-hand smoke. The Kruskal–Wallis test showed a higher concentration of MEP (*p* = 0.022), ∑LMWP (*p* = 0.029) and testosterone (*p* = 0.05) during the early pregnancy in smokers in comparison to non-smokers. Moreover, we found significantly higher birth weight (*p* = 0.036) and head circumference (*p* = 0.048) in smokers’ newborns compared to those of non-smokers.

### 3.5. Analysis of Concentrations of Phthalate Metabolites, and Endocrine Parameters in Association with Newborns’ Sex

When we compared the concentrations of phthalate metabolites and endocrine parameters during the pregnancy, we found significantly higher (*p* ≤ 0.05) concentrations of MiBP, OH-MnBP, ∑DiBP, and ∑DBP, and significantly lower concentrations of FSH, LH, testosterone, and SHBG in the group of mothers with male newborns (Table 7).

### 3.6. Spearman Correlation of Maternal Phthalate Metabolites with Newborns’ Outcomes

Spearman correlation analysis between birth outcomes and phthalate metabolite concentrations in urine samples collected during the early stage of pregnancy (8.–15. week of gestation) and the third trimester of pregnancy detected some positive and negative correlations highlighted in Figure 1 and Figure 2. We also evaluated these correlations separately for both sexes (Figure 3, Figure 4, Figure 5 and Figure 6). We found significant positive correlations between phthalate metabolites and gestational age at delivery, birth weight, and birth length in all newborns as well as in groups of boys and girls (Figure 1, Figure 2, Figure 3, Figure 4, Figure 5 and Figure 6). In the case of head circumference, we found different results based on the sex of newborns. In girls, exposure to phthalates during the early stage of pregnancy (MEP, oxo-MiNP, ΣLMWP; Figure 5) and third trimester (MEP, MiBP, OH-MnBP, OH-MiNP, oxo-MiNP, cx-MiNP, ΣDiBP, ΣDBP, ΣLMWP, ΣHMWP; Figure 6) was positively associated with head circumference. In boys, higher levels of oxo-MiNP during the early stage of pregnancy were negatively associated with head circumference (Figure 3). In the case of Apgar score at 1 min after delivery, we also observed different results based on the sex of newborns. We found that in girls, higher exposure to phthalates during the third trimester (MEP, ΣLMWP; Figure 6) was associated with a higher Apgar score; however, a negative association (MnBP, OH-MEHP, oxo-MEHP, cx-MEPP, ΣDiBP, ΣDBP, ΣDEHP, ΣLMWP; Figure 4) was observed in boys.

### 3.7. Multivariate Regression Models of Birth Outcomes Based on Maternal Endocrine Parameters and Phthalate Metabolites Levels

Using multivariate regression, we examined how phthalate exposure may affect the relationship between maternal endocrine parameters and birth outcomes in neonates (Table 8 and Table 9). We found similar associations when comparing the timing of phthalate exposure during pregnancy. During the early stage and the third trimester of pregnancy, we found a significant relationship between maternal endocrine parameters and gestational age at delivery, birth weight, and head circumference of newborns. However, we observed a significant relationship between maternal SHBG only during early pregnancy, and Apgar score 5 min after the delivery. The statistical significance of hierarchical multivariate regression models increased after adding the concentrations of phthalate metabolites into models. We did not observe any significant relationships containing birth length, ponderal index, or Apgar score at 1 min after the delivery.

## 4. Discussion

A strong relationship between phthalate exposure and disruption of hormonal concentrations in pregnant women has been previously reported (reviewed in [10]). Previous studies have also examined the association between birth outcomes and hormonal balance during pregnancy and relations between prenatal phthalate exposure and birth outcomes [11]. To our knowledge, no study has been conducted to show their mutual relationship. Our data suggest that the maternal endocrine system modulating by phthalate exposure could result in the modification of birth outcomes of newborns.

### 4.1. Biomonitoring of Maternal Phthalate Metabolites

We observed that the concentrations of OH-MEHP, oxo-MEHP, OH-MiNP, oxo-MiNP, ∑DEHP, ∑DiNP, and ∑HMWP in urine samples were significantly higher in the third trimester in comparison to early pregnancy. According to Li et al. [22], phthalate metabolite concentrations throughout the pregnancy were U-shaped; the highest concentrations of phthalates were detected in the 1st and third trimesters. A similar trend to our study was observed in the study of Bustamante-Montes et al. [23]; they noticed higher concentrations of MBP, MEP, and MEHP during the third trimester compared to the first trimester in the cohort of pregnant women. The possible explanation of higher exposure to certain phthalates during the third trimester might be a difference in caloric intake throughout the pregnancy. Early pregnancy does not require a higher caloric intake. However, from the second trimester, caloric demand gradually increases [24], which might lead to a higher risk of phthalate exposure (exceptionally high-molecular-weight phthalates) due to the consumption of contaminated food. Body-fat content also gradually increases during pregnancy, which is associated with higher chronic exposure to phthalates, since fat provides a deposit for lipophilic substances such as phthalates [25].

### 4.2. Analysis of Maternal Baseline Characteristics in Association with Concentrations of Phthalate Metabolites, Endocrine Parameters, and Newborns’ Outcomes

One of the most significant maternal characteristics is maternal age. We found negative associations with ∑DBP, estradiol, and newborns’ birth length. A study by Wu et al. [26] showed that a higher age in pregnant women was associated with decreased levels of phthalates. We hypothesise that older women do not use so many cosmetic products compared to younger women, which might explain lower concentrations of dibutyl phthalate metabolites. In the case of decreased levels of estradiol, the results are inconsistent. According to Schock et al. [27], maternal age was not a significant predictor of hormonal levels. However, the study of Toriola et al. [28] showed that estradiol levels were decreased in older pregnant women. Maternal age has a significant effect on the pregnancy outcome in general. A study by Schimmel et al. [29] reported an increased risk of intrauterine growth restriction of the foetus in older pregnant women.

Maternal education was associated with cortisol levels. Pregnant women with a university or college education had lower levels of cortisol in comparison to high-school-educated women. This can be explained by worse job conditions and more stressful events at work in the case of women with a high school education [30], leading to stress and an increase in cortisol [31]. The maternal living area was associated with levels of TSH. In women living in rural areas, lower concentrations of TSH were observed compared to women living in urban areas. The same pattern was observed in the study of Bojar et al. [32], wherein women from rural areas were observed to have decreased levels of TSH below the reference values. Researchers explained that people from rural areas do not have easy access to the hospital, so they might be undiagnosed for some disorders for a long time. However, in the case of our study, probands living in rural areas had concentrations of TSH within reference values for pregnant women. Therefore, other factors might be related to the TSH and living area, such as different lifestyles [32].

The next significant maternal characteristic was BMI. Maternal BMI was positively associated with some of the phthalate metabolites. This association has been well documented in other studies (reviewed in [33]). It can be explained by phthalates’ lipophilic properties, leading to chronic internal exposure [8]. Moreover, maternal BMI was negatively associated with SHBG. Our results are in agreement with the study by Akin et al. [34], who showed that obesity in women was associated with lower SHBG.

Exposure to tobacco smoke and second-hand smoke are significant factors related to maternal phthalate exposure. We observed that probands who were smokers during pregnancy had higher concentrations of MEP and ∑LMWP. Surprisingly, exposure to second-hand smoke was positively associated with more phthalate metabolites (MEP, MiBP, MnBP, OH-MiBP, OH-MnBP, MBzP, cx-MiNP, ∑DiBP, ∑DnBP, ∑DBP, ∑LMWP). Our results are in agreement with other studies, because active and passive smoking are considered as the potential source of phthalate exposure [35]. However, exposure to tobacco smoke and second-hand smoke might be risk factors for maternal and neonatal health as well. Our study showed that smokers had significantly higher concentrations of testosterone, which is in accordance with the study of Toriola et al. [28]. Moreover, in the group of newborns whose mothers were smokers or exposed to second-hand smoke, higher gestational age, birth weight, and head circumference were noted. Our results are surprising since most studies showed that smoking is inversely associated with gestational age or weight of newborns [36,37,38].

We also evaluated the health status of our probands in association with phthalate metabolites, endocrine parameters, and neonatal outcomes. We found significantly higher levels of OH-MiBP in women who used pharmaceuticals. Many studies showed that pharmaceuticals could be a source of phthalate exposure [39,40]. When we analysed the specific type of pharmaceuticals—hormonal therapy—we found surprisingly higher levels of MnBP, OH-MEHP, and oxo-MEHP in women who did not use hormonal therapy, which is in disagreement with the studies of Al Saleh et al. [41,42], who observed increased phthalate metabolite concentrations in probands undergoing in vitro fertilisation [41], and women with contraceptive-use history [42]. The possible explanation for this inconsistency may be that in our study, we evaluated the use of hormonal therapy in general, regardless of its division into contraceptives, in vitro-fertilisation pharmaceuticals, and other groups of treatment, leading to possible different results. In the case of endocrine parameters, using pharmaceuticals and hormonal therapy was associated with increased levels of FT4, and decreased levels of TSH, FT3 and cortisol, respectively, which can be explained by the interaction of pharmaceuticals with the endocrine system at multiple different levels leading to altered concentrations of hormones [43]. In addition, we observed higher levels of MMP, SHBG in women diagnosed with a thyroid disorder (hypothyroidism or thyroid cyst) compared to women without thyroid disorder. Maternal thyroid disorder was also positively associated with newborns’ birth length and ponderal index. Lower levels of FT4 and cortisol levels were observed in women diagnosed with ovarian disorders (ovarian cysts). The proper function of the thyroid gland and gonads is ensured primarily by balance within the endocrine system. Within this system, the connection of several hormones between the axes of the hypothalamus, pituitary, thyroid, gonads, or adrenal glands was monitored. Therefore, we can hypothesise that a failure in one of these axes can cause a change in the others [44]. It is vital to notice that maternal thyroid hormones are essential for proper foetal development. Many studies reported the association between maternal thyroid disorders and poor neonatal outcomes [5]. Surprisingly, our results showed that newborns of women with diagnosed thyroid disorder had a higher ponderal index and birth length. We can hypothesise that maternal hypothyroidism did not affect the intrauterine development of the newborn due to the use of hormone replacement therapy [45]. The thyroid function can also be impaired by phthalates, substances with a similar chemical structure to those of hormones. Several studies observed the associations between phthalate exposure and altered thyroid hormone levels, potentially leading to disorder (reviewed in [10]). However, we cannot draw conclusions based on these associations due to the low number of probands with ovarian (n = 2) and thyroid disorders (n = 8).

### 4.3. Analysis of Concentrations of Phthalate Metabolites, and Endocrine Parameters in Association with Newborns’ Sex

When we compared the concentrations of phthalate metabolites and endocrine parameters during the pregnancy, we found significantly higher concentrations of MiBP, OH-MnBP, ∑DiBP, and ∑DBP, and significantly lower concentrations of FSH, LH, testosterone, and SHBG in the group of mothers with male newborns. The possible explanation for increased levels of phthalate metabolites in mothers with male newborns might be due to the higher caloric intake in pregnancy with male foetuses compared to pregnancy with female foetuses [46]. Therefore, it might lead to a higher risk of phthalate exposure due to the consumption of contaminated food. In the case of endocrine parameters, pregnant women with female foetuses had slightly higher concentrations of certain hormones which is in accordance with other studies [28,47]. However, maternal testosterone levels tend to be higher in women pregnant with male foetuses, according to other studies [48], which is in disagreement with our results.

### 4.4. Associations between Maternal Phthalate Metabolites and Birth Outcomes

In our cohort, we mainly observed positive correlations between the concentrations of phthalate metabolites and birth outcomes, e.g., gestational age, birth weight, and birth length. However, in the case of head circumference and the Apgar score, we observed some discrepancies. The results of epidemiological studies analysing the relationship between prenatal phthalate exposure and birth outcomes are inconsistent. As in our cohort, other epidemiological studies have shown a positive relationship between concentrations of phthalate metabolites, e.g., MCPP [49], ∑DEHP [50], and ∑LMWP [51] in urine samples of pregnant women and gestational age of newborns of both sexes. Metabolites MBP, MEHP [52], oxo-MiNP [53], ∑DnBP [54], MCOP, and ∑DEHP [55] during pregnancy were positively associated with neonatal birth weight, especially in boys. The association between ∑DnBP throughout pregnancy and boys’ head circumference was positive [54]. MCPP concentration was positively associated with girls’ birth length, weight, and head circumference [54]. However, several studies have observed negative associations between prenatal phthalate exposure and birth outcomes. Elevated concentrations of MBP, MEHP [56], MEP [57], ∑DnBP [54] in pregnant women were associated with a lower gestational age of newborns of both sexes. Concentrations of oxo-MiNP [57] and ∑DnBP [53] were inversely associated with neonatal head circumference. Some studies noticed the negative associations between maternal concentrations of MMP, MEP [52], MBP, MEHP [56], OH-MEHP, oxo-MEHP [52], ∑DnBP, ∑DEHP [58], and neonatal birth weight. Maternal exposure to DEHP during pregnancy was associated with reduced foetal growth [59]. The Chinese study by Gao et al. [60] showed that the cumulative risk index of phthalate exposure (DEP, DBP, BBzP, and DEHP) in pregnant women was associated with reduced birth weight and length of newborns of both sexes. Few studies have examined the relationship between prenatal phthalate exposure and the Apgar score. Two epidemiological studies did not find any significant relationship [61,62], but Li et al. [63] reported significantly lower Apgar scores in neonates with higher prenatal phthalate exposure. Inconsistent results of the association between prenatal phthalate exposure and birth outcomes may be due to phthalates’ different mechanisms of action during prenatal development. Phthalates can alter birth outcomes through epigenetic processes [49], inflammatory processes [64], or oxidative stress [65], leading to a wide range of changes in birth outcomes. Moreover, when it comes to sex differences in neonatal outcomes, it might be explained by different concentrations of phthalate metabolites and endocrine parameters of female and male newborns’ mothers, as was discussed in the previous section. We can speculate that this could lead to altered neonatal outcomes in a different manner depending on the sex of the newborn.

### 4.5. Relationships between Phthalate Metabolites Levels, Maternal Endocrine Parameters, and Birth Outcomes

We hypothesise that phthalates as endocrine disruptors may affect birth outcomes through interaction with the maternal endocrine system. It is known that phthalates in the body interface with the endocrine system at several levels, i.e., at the level of the hypothalamus; pituitary gland; peripheral organs, such as ovary, testis, thyroid, or adrenal glands; transport proteins; and most significantly, phthalates can interact with nuclear receptors (reviewed in [10]). Therefore, we decided to analyse the mutual relationship between phthalate metabolites and endocrine parameters in our cohort using hierarchical multivariate regression.

We found that the Apgar score 5 min after delivery was positively associated with the concentration of maternal SHBG during early pregnancy, with increasing statistical significance after the addition of MBzP to this model. To our knowledge, no studies show the relationship between SHBG and Apgar score. However, the study observed a positive association between maternal SHBG levels during the 26th week of pregnancy and foetal growth restriction [66], leading to lower Apgar scores and neonatal morbidity [67], which contradicts our results. In the relationship between maternal SHBG and neonatal health, the trimester of pregnancy should be considered when SHBG has been analysed, as SHBG, as with estradiol, increases during pregnancy [68], which may be the factor for discrepancy mentioned above. We further found that gestational age was positively associated with maternal SHBG and FSH levels during early pregnancy. These associations were more significant after the addition of MBzP to the model. The study by Valdés et al. [69] and Xargay-Torrent et al. [66] did not show any significant association between SHBG levels and gestational age of the neonate. The positive association between gestational age and FSH is a question to arouse interest because its concentration remains very low during pregnancy. It is often not possible to quantify these concentrations from the blood of a pregnant woman. This is because estradiol levels increase several times during pregnancy and therefore inhibit gonadotropin secretion by negative feedback [70]. Research has shown that FSH is essential for maintaining pregnancy and timing childbirth [71]. However, it is difficult to draw conclusions from our data. We can only hypothesise because we monitored FSH levels during the early stage of pregnancy and at the beginning of the third trimester of pregnancy, not immediately before baby delivery. In the case of the relationship between birth weight and estradiol level in the group of boys and the whole cohort of children, a positive association was reported, while it was more significant after the addition of MBzP and OH-MnBP to the models. Our results agree with several studies showing the positive association of maternal estradiol level throughout pregnancy with neonatal weight [3,72,73]. Moreover, Zhang et al. [73] noted this positive relationship in newborn boys. However, at very high maternal estradiol concentrations, growth retardation may occur in the foetus [74]. We also observed a positive association of maternal FT4 concentrations during the early stage of pregnancy with weight in female newborns. This relationship was statistically more significant after the addition of MBzP to this model. Besides, we observed a negative association between maternal FT4 level and head circumference in boys, where the relationship was more significant after the addition of oxo-MiNP to this model. In general, there is evidence of a positive relationship between maternal thyroid hormonal levels and foetal growth [5], as suggested by our association between FT4 levels and weight in female newborns. However, in a condition where a pregnant woman is deficient in thyroid hormones, an increase in head circumference may occur in the foetus due to decreased thyroid hormones [75], as suggested by our results, as well.

Analysing the relationship between birth outcomes and maternal endocrine parameters during the third trimester, we observed a positive relationship between the gestational age of female newborns and maternal TSH concentrations, with increasing statistical significance after the addition of OH-MiBP to the model. This association is a question that arouses interest. Higher TSH concentrations during pregnancy are associated with the risk of preterm birth or miscarriage, depending on the period of pregnancy [76]. Therefore, further research is needed to explain these discrepancies. We also noted a negative relationship between head circumference in girls with maternal FSH level, increasing statistical significance after adding cx-MiNP, OH-MnBP, and oxo-MEHP to the model. We hypothesise that there may be negative feedback within the HPG axis in pregnant women with very high estradiol levels, leading to reduced FSH secretion [70]. Because decreased FSH concentrations are associated with high estradiol concentrations during pregnancy, this relationship may result in potential growth retardation of the foetus, which may be reflected in reduced head circumference in neonates [74]. The study by Chellakooty et al. [77] reported a similar result: higher FSH concentrations in newborn females were associated with lower birth weight, length, and head circumference [77]. However, this relationship was not profoundly studied because FSH concentration remains low during pregnancy, often on the quantification threshold. Therefore, further research is needed to explain the association between FSH and neonatal head circumference. We also reported the associations between gestational age, estradiol (positive association), and FT4 (negative association). In the group of boys, we also noted a positive association between gestational age and estradiol level and increased significance after adding MBzP and MEP metabolites to the models. The positive association between estradiol level and gestational age of the neonate is consistent with other studies, as maternal estradiol concentration also increases with increasing gestational weeks [78]. The negative association of gestational age and FT4 level observed in our study is consistent with the reference values for pregnant women [79]. During early stage of pregnancy, the foetus is wholly dependent on the mother’s thyroid hormones. As pregnancy progresses, the foetus begins to produce hormones on its own [80], which may explain the decline in maternal hormones during pregnancy.

Our results of hierarchical multivariate regression suggest that phthalates may affect newborn´s health also through maternal endocrine parameters. Phthalate metabolites MEP, OH-MnBP, MBzP, and cx-MiNP, as with maternal estradiol, were in positive association with gestational age or neonatal birth weight, suggesting potential estrogenic activity of selected phthalate metabolites, as confirmed by many in vitro studies (reviewed in [81]). In addition, MBzP, as with SHBG, was in positive association with gestational age, which also suggests the estrogenic activity of this phthalate metabolite. In the case of associations with FSH and FT4 hormones, the direction of association (positive/negative) depended on neonatal markers and the specificity of phthalate metabolites. As an example, we can show a model where FSH was in negative association with head circumference in girls, with oxo-MEHP acting in the same direction as FSH. In contrast, the metabolites OH-MnBP and cx-MiNP were in positive association with this neonatal marker, thus acting in opposition to FSH. Therefore, it is important to emphasise the specificity of phthalate metabolites, which depends on, for example, the molecular weight of phthalate metabolites [82].

## 5. Strengths and Limitations of Study

The strength of our study is the sample-collection timing. We collected urine and blood samples during the early stage of pregnancy, which is the crucial developmental period for the embryo and foetus. It allows us to examine the maternal endocrine parameters and phthalate exposure related to newborns’ health-related outcomes in our future longitudinal study. Moreover, so far, no study has shown the mutual relationship between maternal endocrine parameters and phthalate exposure on the disruption of newborn outcomes. We are the first to show this mutual relationship. While our results may have crucial public health significance, the current study has several limitations. The weakness is the size of a cohort of 72 mother–newborn pairs. The second limitation is that some of the questionnaires were not fully completed.

## 6. Conclusions

In summary, it is necessary to monitor maternal hormones, especially during the first trimester of pregnancy, when foetal programming for organ-system development occurs. We are the first to show that prenatal phthalate exposure may modulate the effects of maternal endocrine parameters on newborns’ birth outcomes. Further research is needed to examine the relationship between birth outcomes and endocrine parameters during pregnancy concerning maternal phthalate exposure in larger sample sizes that can accurately assess those effects in newborns.

## Figures and Tables

**Figure 1 children-09-00413-f001:**
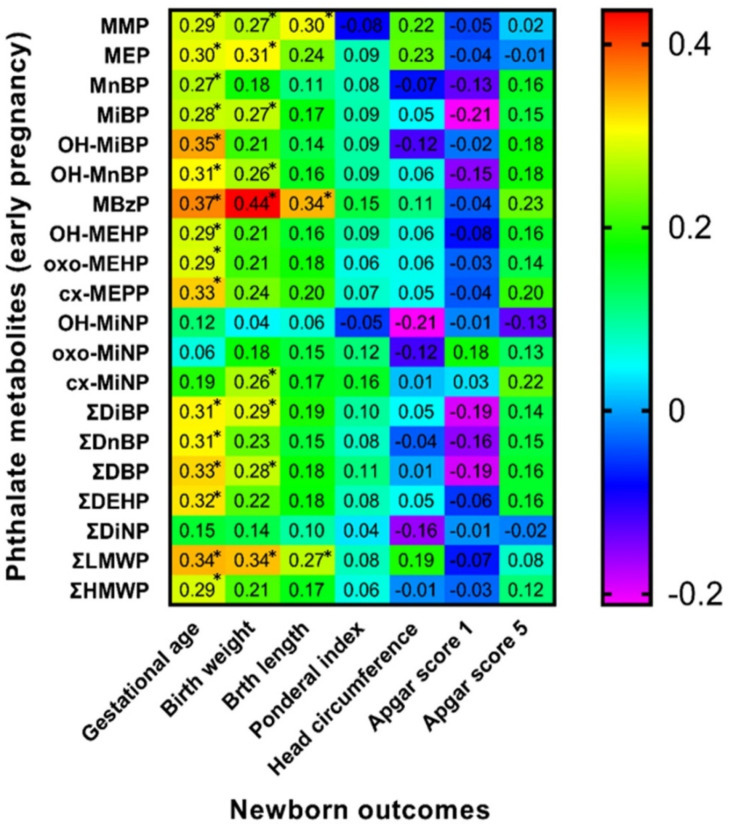
Correlation matrix of newborn outcomes and maternal phthalate metabolites during early pregnancy (≤15. w.g.). Legend is shown in the footnote of Table 4. * The correlation is significant at the level *p* ≤ 0.05.

**Figure 2 children-09-00413-f002:**
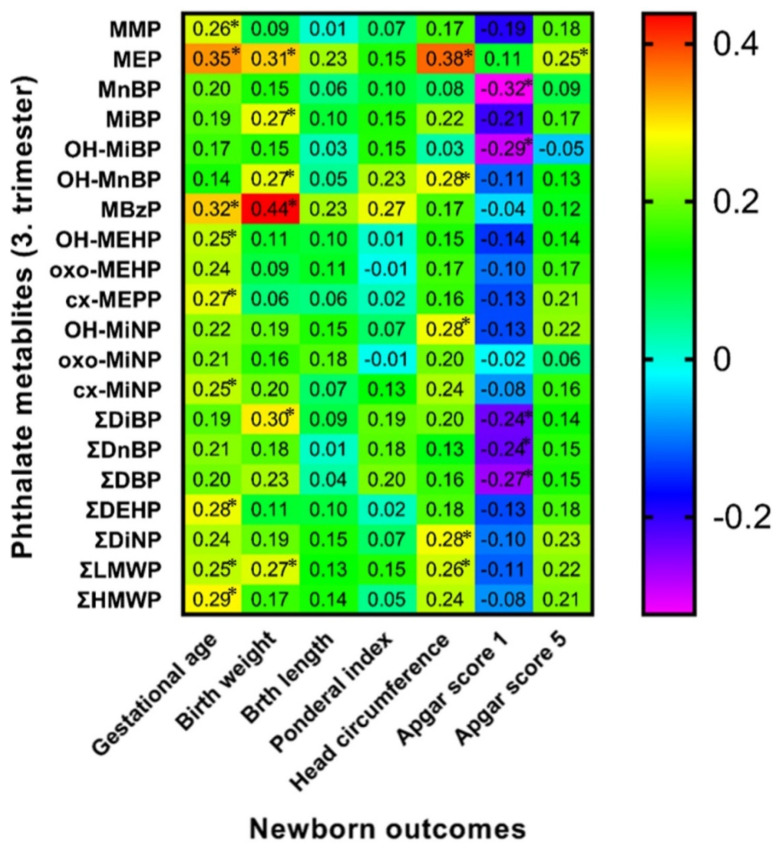
Correlation matrix of newborn outcomes and maternal phthalate metabolites during third trimester of pregnancy. Legend is shown in the footnote of Table 4. * The correlation is significant at the level *p* ≤ 0.05.

**Figure 3 children-09-00413-f003:**
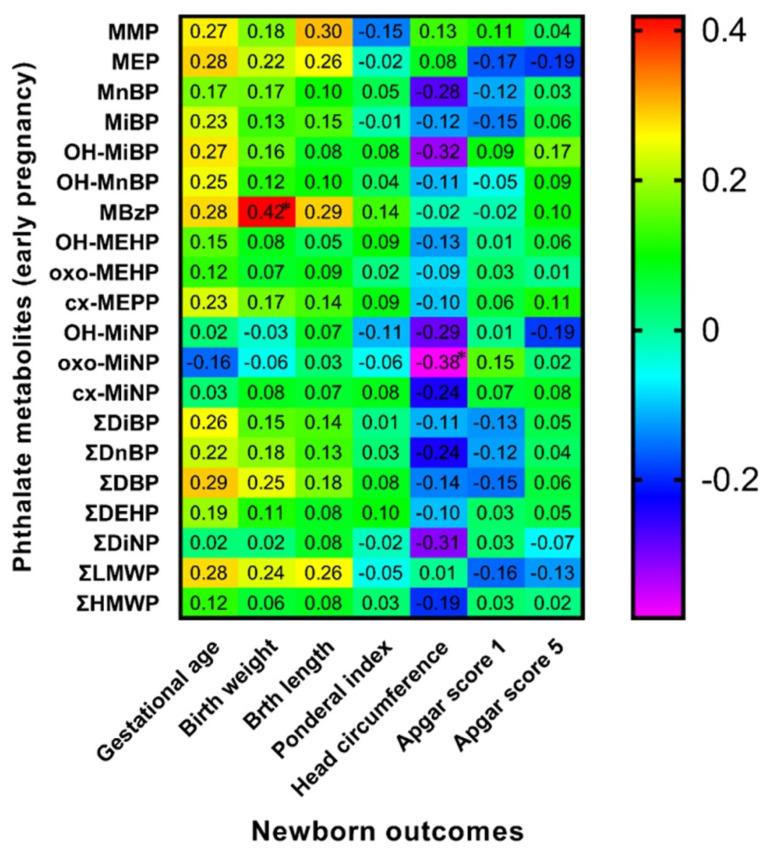
Correlation matrix of boy-newborn outcomes and maternal phthalate metabolites during early pregnancy (≤15. w.g.). Legend is shown in the footnote of Table 4. * The correlation is significant at the level *p* ≤ 0.05.

**Figure 4 children-09-00413-f004:**
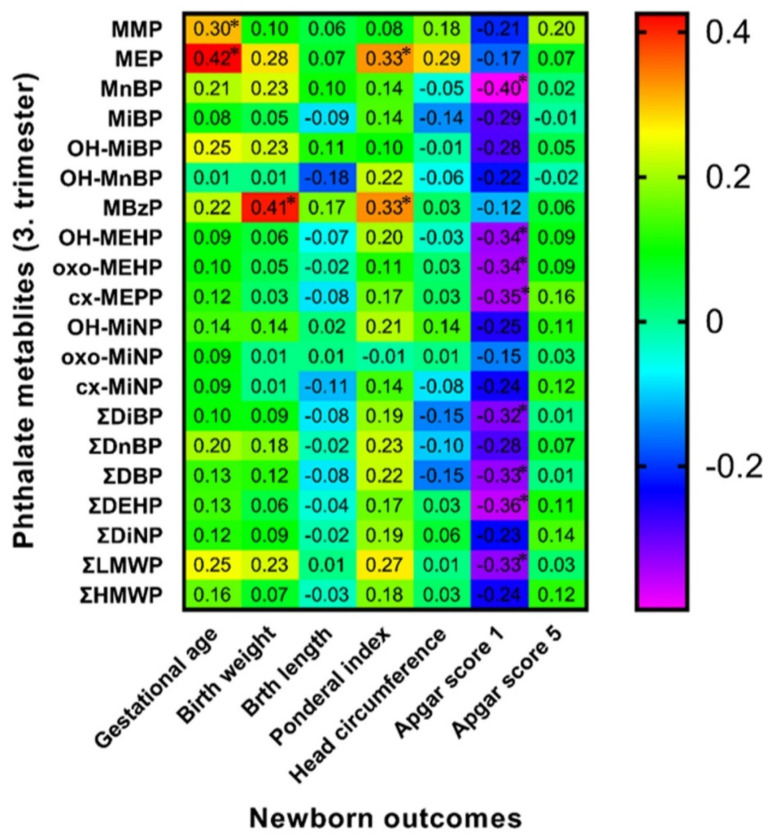
Correlation matrix of boy-newborn outcomes and maternal phthalate metabolites during third trimester of pregnancy. Legend is shown in the footnote of Table 4. * The correlation is significant at the level *p* ≤ 0.05.

**Figure 5 children-09-00413-f005:**
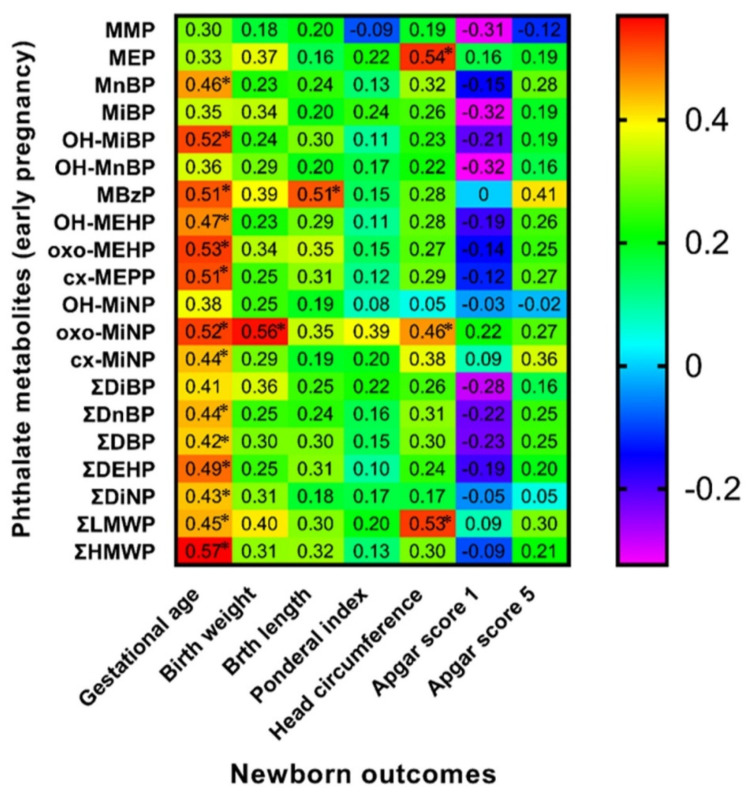
Correlation matrix of girl-newborn outcomes and maternal phthalate metabolites during early pregnancy (≤15. w.g.). Legend is shown in the footnote of Table 4. * The correlation is significant at the level *p* ≤ 0.05.

**Figure 6 children-09-00413-f006:**
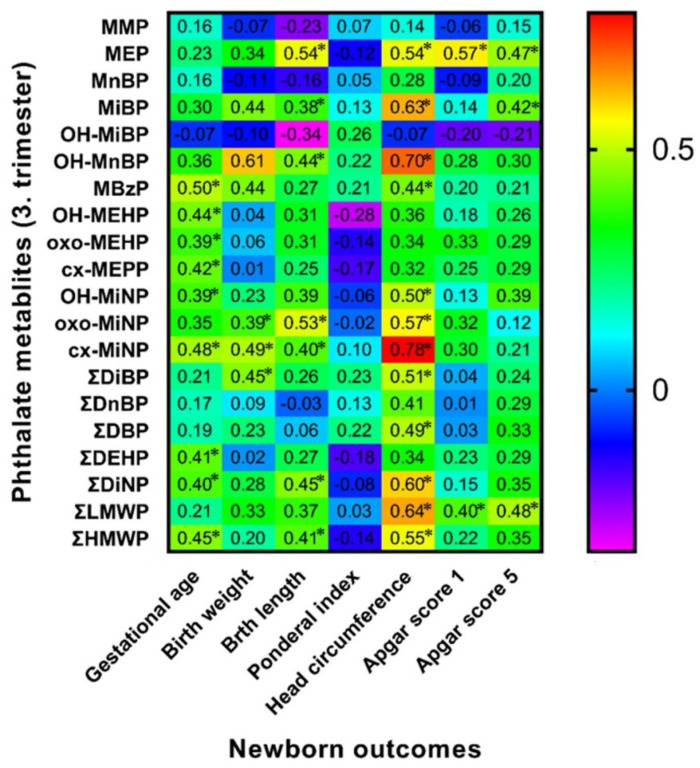
Correlation matrix of girl-newborn outcomes and maternal phthalate metabolites during third trimester of pregnancy. Legend is shown in the footnote of Table 4. * The correlation is significant at the level *p* ≤ 0.5.

**Table 1 children-09-00413-t001:** Apgar score.

Parameter	Points		
	0	1	2
Number of heartbeats	Absence	<100	>100
Respiration	Absence	Hypoventilation	Strong crying
Muscle tonus	Weak	Some flexions	Active movement
Reflexes	No reaction	Grimace	Sneezing, coughing
Colour	Blue, white	Pink body, blue peripheral parts of the body	Pink

Total score: 9–10 points: normal, 5–8 points: danger, <5: life danger.

**Table 2 children-09-00413-t002:** Characteristics of mothers of our study.

Parameter	Mean ± SD/Frequency (n; %)					
Age	31.91 ± 5.23					
BMI	24.25 ± 4.79					
Total score in consumer practices (early pregnancy)	11.09 ± 2.96					
Total score in consumer practices (third trimester of pregnancy)	11.7 ± 3.37					
Parity	0.83 ± 0.97					
	zero (n = 34; 48.57%)	one (n = 20; 28.57%)	two (n = 12; 17.14%)		three (n = 3; 4.29%)	four (n = 1; 1.43%)
Education	high school (n = 23; 32.86%)			college/university (n = 47; 67.14%)		
Living area	rural (n = 34; 48.57%)			urban (n = 36; 51.43%)		
Smoker	non-smoker (n = 48; 68.57%)		former smoker (n = 18; 25.71%)		smoker (n = 4; 5.71%)	
Second-hand smoke	no (n = 50; 78.57%)			yes (n = 15; 21.43%)		
Using pharmaceuticals	no (n = 40; 62.5%)			yes (n = 24; 37.5%)		
Hormonal therapy	no (n = 29; 43.28%)			yes (n = 38; 56.72%)		
Thyroid disorder	no (n = 62; 88.57%)			yes (n = 8; 11.43%)		
Ovarian disorder	no (n = 68; 97.14%)			yes (n = 2; 2.86%)		

Legend: n—number, SD—standard deviation.

**Table 3 children-09-00413-t003:** Characteristics of newborns’ outcomes.

**All Newborns’ Outcomes (n = 72)**								
	**Gestational Age at Delivery (weeks)**	**Birth Weight (g)**	**Birth Length (cm)**	**Ponderal Index**	**Head Circumference (cm)**	**Apgar Score 1**	**Apgar Score 5**	**Apgar Score 10**
MEAN	38.99	3333.89	49.74	27.03	34.47	9.17	9.83	10.00
MIN	31.00	1900.00	44.00	20.53	30.00	1.00	5.00	10.00
MED	39.00	3355.00	50.00	27.07	34.00	9.00	10.00	10.00
MAX	42.00	4420.00	55.00	33.23	39.00	10.00	10.00	10.00
SD	1.64	499.93	2.30	2.94	1.71	1.39	0.65	0.00
**Boys’ Outcomes (n = 44)**								
	**Gestational Age at Delivery (weeks)**	**Birth Weight (g)**	**Birth Length (cm)**	**Ponderal Index**	**Head Circumference (cm)**	**Apgar Score 1**	**Apgar Score 5**	**Apgar Score 10**
MEAN	39.14	3421.14	50.14	27.09	34.74	9.14	9.91	10.00
MIN	35.00	2270.00	46.00	20.53	30.00	4.00	9.00	10.00
MED	39.00	3460.00	50.00	27.33	34.50	9.00	10.00	10.00
MAX	42.00	4420.00	55.00	33.23	39.00	10.00	10.00	10.00
SD	1.47	496.30	2.32	2.83	1.84	1.17	0.29	0.00
**Girls’ Outcomes (n = 28)**								
	**Gestational Age at Delivery (weeks)**	**Birth Weight (g)**	**Birth Length (cm)**	**Ponderal Index**	**Head Circumference (cm)**	**Apgar Score 1**	**Apgar Score 5**	**Apgar Score 10**
MEAN	38.75	3196.79	49.11	26.93	34.04	9.21	9.71	10.00
MIN	31.00	1900.00	44.00	20.85	31.00	1.00	5.00	10.00
MED	39.00	3240.00	50.00	26.45	34.00	10.00	10.00	10.00
MAX	41.00	4140.00	53.00	33.12	36.50	10.00	10.00	10.00
SD	1.88	482.74	2.17	3.14	1.41	1.71	0.98	0.00

Legend: MAX—maximum, MED—median, MIN—minimum, SD—standard deviation.

**Table 4 children-09-00413-t004:** Descriptive statistics of phthalate metabolite concentrations (ng/mL).

	Early Pregnancy					Third Trimester of Pregnancy				
Compound Name	% ≥ LOQ	MIN	MED	MAX	SD	% ≥ LOQ	MIN	MED	MAX	SD
MMP	30.51	≤LOQ	≤LOQ	26.21	4.16	24.29	≤LOQ	≤LOQ	30.09	5.53
MEP	91.53	≤LOQ	20.64	629.28	92.94	95.72	≤LOQ	18.50	874.13	126.11
MiBP	90.00	≤LOQ	14.04	420.07	46.81	100.00	≤LOQ	15.00	2615.50	268.56
MnBP	88.33	≤LOQ	17.17	415.46	46.90	92.86	≤LOQ	17.07	161.90	27.92
OH-MiBP	71.67	≤LOQ	1.89	38.61	4.50	81.43	≤LOQ	2.02	18.39	2.75
OH-MnBP	91.67	≤LOQ	7.03	132.97	15.85	97.14	≤LOQ	7.58	643.97	66.81
MnPeP	0.00	≤LOQ	≤LOQ	≤LOQ	0.00	0.00	≤LOQ	≤LOQ	≤LOQ	0.00
MCHP	0.00	≤LOQ	≤LOQ	≤LOQ	0.00	0.00	≤LOQ	≤LOQ	≤LOQ	0.00
MBzP	50.00	≤LOQ	1.17	28.72	3.34	35.71	≤LOQ	≤LOQ	39.76	4.55
OH-MEHP	81.67	≤LOQ	4.02	43.20	6.72	94.29	≤LOQ	4.93	110.30	15.71
oxo-MEHP	83.33	≤LOQ	3.30	25.73	5.07	98.57	≤LOQ	3.82	43.99	6.54
cx-MEPP	96.67	≤LOQ	5.74	53.07	8.80	98.57	≤LOQ	6.74	70.70	9.95
MiNP	0.00	≤LOQ	≤LOQ	≤LOQ	0.00	5.71	≤LOQ	≤LOQ	2.11	0.28
OH-MiNP	76.67	≤LOQ	4.16	82.25	12.13	97.14	≤LOQ	8.18	265.64	7.85
oxo-MiNP	25	≤LOQ	≤LOQ	63.41	6.53	50.00	≤LOQ	≤LOQ	13.03	2.28
cx-MiNP	70	≤LOQ	2.12	151.04	15.67	85.71	≤LOQ	2.54	111.58	11.84
MnOP	0.00	≤LOQ	≤LOQ	≤LOQ	0.00	0.00	≤LOQ	≤LOQ	≤LOQ	0.00
∑DiBP		2.27	16.96	430.17	48.77		2.27	16.70	2616.00	268.56
∑DnBP		2.47	25.45	453.19	55.31		2.47	26.28	645.74	73.05
∑DBP		4.74	44.87	641.98	94.69		4.74	42.17	3261.74	268.56
∑DEHP		2.10	11.48	122.00	20.33		2.10	15.19	162.34	28.26
∑DiNP		2.31	8.09	296.71	31.55		2.51	12.63	270.91	17.79
∑LMWP		8.26	81.14	649.75	132.66		9.29	79.76	3477.38	383.74
∑HMWP		4.41	23.26	324.94	41.09		5.36	32.42	292.70	39.37

Legend: cx-MEPP—mono(2-ethyl-5-carboxypentyl) phthalate, cx-MiNP—mono(carboxy-methyl-heptyl) phthalate, MAX—maximum, MBzP—monobenzyl phthalate, MCHP—monocyclohexyl phthalate, MED—median, MEP—monoethyl phthalate, MiBP—mono-iso-butyl phthalate, MIN—minimum, MiNP—mono-iso-nonyl phthalate, MMP—monomethyl phthalate, MnBP—mono-n-butyl phthalate, n—number of probands, MnOP—mono-n-octyl phthalate, MnPeP—mono-n-pentyl phthalate, OH-MEHP—mono(2-ethyl-5-hydroxyhexyl) phthalate, OH-MiBP—mono(hydroxy-iso-butyl) phthalate, OH-MiNP—mono(hydroxyl-methyl-octyl) phthalate, OH-MnBP—mono(hydroxy-n-butyl) phthalate, oxo-MEHP—mono(2-ethyl-5-oxohexyl) phthalate, oxo-MiNP—mono(oxo-methyl-octyl) phthalate, SD—standard deviation, ∑DBP—sum of dibutyl phthalate metabolites (DnBP+DiBP), ∑DEHP—sum of the secondary metabolites of di(2-ethylhexyl) phthalate (cx-MEPP+OH-MEHP+oxo-MEHP), ∑DiBP—sum of metabolites of di-iso-butyl phthalate (MiBP+OH-MiBP), ∑DiNP—sum of secondary metabolites of di-iso-nonyl phthalate (cx-MiNP+OH-MiNP+oxo-MiNP), ∑DnBP—sum of metabolites of di-n-butyl phthalate (MnBP+OH-MnBP), ∑HMWP—sum of high-molecular-weight phthalate metabolites (DEHP+DiNP), ∑LMWP—sum of low-molecular-weight phthalate metabolites (MMP+MEP+MBzP+DBP), % ≥ LOQ—% of samples above the limit of quantification, ≤LOQ—values below the limit of quantification.

**Table 5 children-09-00413-t005:** Comparison of phthalate concentrations between early pregnancy and third trimester of pregnancy.

		Mean Rank	Mann-Whitney U	Z	*p*
OH-MEHP	Early pregnancy	58.44	1670.00	−2.397	0.017
	Third trimester of pregnancy	74.48			
oxo-MEHP	Early pregnancy	58.44	1670.00	−2.369	0.017
	Third trimester of pregnancy	74.48			
OH-MiNP	Early pregnancy	51.56	1244.00	−4.32	≤0.001
	Third trimester of pregnancy	80.48			
oxo-MiNP	Early pregnancy	57.97	1641.00	−2.903	0.004
	Third trimester of pregnancy	74.89			
∑DEHP	Early pregnancy	58.90	1698.50	−2.266	0.023
	Third trimester of pregnancy	74.08			
∑DiNP	Early pregnancy	51.11	1216.00	−4.444	≤0.001
	Third trimester of pregnancy	80.87			
∑HMWP	Early pregnancy	55.27	1474.00	−3.279	0.001
	Third trimester of pregnancy	77.24			

Legends was shown in the footnote of Table 4.

**Table 6 children-09-00413-t006:** Descriptive statistics of serum hormone concentrations.

		Early Pregnancy				Third Trimester of Pregnancy			
Compound Name	Units	MIN	MED	MAX	SD	MIN	MED	MAX	SD
FSH	mIU/mL	0.04	0.00	0.01	0.43	0.08	0.00	0.05	0.91
LH		0.95	0.09	0.93	2.12	0.28	0.01	0.23	1.11
estradiol	pg/mL	4015.66	851	3079	20,800	26,310.69	3784	27,220	46,233
testosterone	ng/mL	0.75	0.16	0.68	1.51	0.95	0.42	0.89	2.78
TSH	μIU/mL	1.69	0.03	1.44	4.48	2.02	0.40	1.84	5.89
FT3	pg/mL	5.28	0.83	3.62	12.70	6.01	2.49	6.90	12.50
FT4	ng/mL	1.69	0.52	0.86	3.82	2.18	0.47	2.80	3.78
cortisol	μg/mL	86.01	0.85	18.93	412.40	210.70	17.73	266.40	487.20
SHBG	mmol/L	307.98	86.50	306.00	643.20	528.27	235.50	508.10	750

Legend: FSH—follicle-stimulating hormone, FT3—free triiodothyronine, FT4—free thyroxine, LH—luteinising hormone, MAX—maximum, MED—median, MIN—minimum, n—number of probands, SD—standard deviation, SHBG-sex hormone-binding globulin, TSH—thyroid-stimulating hormone.

**Table 7 children-09-00413-t007:** Comparison of phthalate concentrations between mothers with male and female newborns.

Name of Compound	Newborn’s Sex	Mean Rank	Mann–Whitney U	Z	*p*
MiBP	boys	40.53	364	−2.612	0.009
	girls	27.48			
OH-MnBP	boys	39.73	398.5	−2.196	0.028
	girls	28.76			
∑DiBP	boys	40.77	354	−2.733	0.006
	girls	27.11			
∑DBP	boys	39.83	394.5	−2.244	0.025
	girls	28.61			
FSH	boys	28.22	296	−3.074	0.002
	girls	43.12			
LH	boys	27.32	259	−3.527	≤0.001
	girls	44.54			
testosterone	boys	28.59	311	−2.857	0.004
	girls	42.54			
SHBG	boys	29.75	370	−1.969	0.049
	girls	39.27			

Legends are shown in the footnotes of Table 4 and Table 6.

**Table 8 children-09-00413-t008:** Relationship between neonatal parameters and maternal endocrine parameters (early pregnancy) in the context of phthalate exposure.

Model: Gestational Age at Delivery		B (SE)	β (95% CI)	*p* (Coefficient)	r^2^ (Model)	*p* (Model)
1	SHBG	2.954 (1.288)	0.288 (0.376; 5.531)	0.025	0.083	0.025
2	SHBG	3.165 (1.246)	0.309 (0.671; 5.660)	0.014	0.161	0.007
	FSH	0.616 (0.267)	0.280 (0.081; 1.151)	0.025		
3	SHBG	3.692 (1.219)	0.360 (1.250; 6.134)	0.004	0.002	0.237
	FSH	0.561 (0.258)	0.255 (0.044; 1.078)	0.034		
	MBzP	1.126 (0.476)	0.282 (0.172; 2.080)	0.022		
**Model: Apgar score 5**		**B (SE)**	**β (95% CI)**	** *p* ** **(coefficient)**	**r^2^ (model)**	** *p* ** **(model)**
1	SHBG	1.405 (0.541)	0.325 (0.322; 2.487)	0.012	0.106	0.012
2	SHBG	1.612 (0.535)	0.373 (0.540; 2.683)	0.004	0.170	0.005
	MBzP	0.435 (0.210)	0.257 (0.016; 0.855)	0.042		
**Model: Head circumference in boys**		**B (SE)**	**β (95% CI)**	** *p* ** **(coefficient)**	**r^2^ (model)**	** *p* ** **(model)**
1	FT4	−2.284 (1.027)	−0.361 (−4.373; −0.194)	0.033	0.130	0.033
2	FT4	−2.167 (0.975)	−0.343 (−4.153; −0.181)	0.033	0.242	0.012
	oxo-MiNP	−2.051 (0.944)	−0.335 (−3.974; −0.128)	0.037		
**Model: Birth weight in girls**		**B (SE)**	**β (95% CI)**	** *p* ** **(coefficient)**	**r^2^ (model)**	** *p* ** **(model)**
1	FT4	708.783 (324.157)	0.431 (34.661; 1382.906)	0.040	0.185	0.040
2	FT4	684.951 (282.088)	0.416 (96.525; 1273.377)	0.025	0.413	0.005
	MBzP	664.526 (238.608)	0.477 (166.797; 1162.254)	0.011		

Legend: β—standardised coefficient, B—unstandardised coefficient, CI—confidence interval, FSH—follicle-stimulating hormone, FT4—thyroxin, MBzP—monobenzyl phthalate, oxo-MiNP—mono(oxo-methyl-octyl) phthalate, *p*—statistical significance value, r^2^—coefficient of determination, SE—standard error, SHBG—sex hormone-binding globulin.

**Table 9 children-09-00413-t009:** Relationship between neonatal parameters and maternal endocrine parameters (third trimester of pregnancy) in the context of phthalate exposure.

Model: Gestational Age at Delivery		B (SE)	β (95% CI)	*p* (Coefficient)	r^2^ (Model)	*p* (Model)
1	Estradiol	3.294 (0.879)	0.424 (1.537; 5.050)	<0.001	0.180	<0.001
2	Estradiol	3.166 (0.851)	0.408 (1.465; 4.867)	<0.001	0.246	<0.001
	FT4	−1.006 (0.426)	−0.259 (−1.858; −0.154)	0.021		
3	Estradiol	3.004 (0.826)	0.387 (1.353; 4.654)	0.001	0.307	<0.001
	FT4	−1.022 (0.412)	−0.263 (−1.846; −0.198)	0.016		
	cx-MiNP	0.926 (0.397)	0.248 (0.133; 1.720)	0.023		
**Model: Birth weight**		**B (SE)**	**β (95% CI)**	** *p* ** **(coefficient)**	**r^2^ (model)**	** *p* ** **(model)**
1	Estradiol	716.154 (329.426)	0.262 (58.050; 1374.258)	0.033	0.069	0.033
2	Estradiol	613.881 (305.844)	0.225 (2.700; 1225.063)	0.049	0.217	<0.001
	MBzP	397.493 (114.982)	0.387 (167.721; 627.266)	0.001		
3	Estradiol	766.434 (307.741)	0.281 (151.268; 1381.599)	0.015	0.266	<0.001
	MBzP	288.631 (124.299)	0.281 (40.161; 537.101)	0.024		
	OH-MnBP	302.944 (148.777)	0.250 (5.542; 600.346)	0.046		
**Model: Gestational age at delivery in boys**		**B (SE)**	**β (95% CI)**	** *p* ** **(coefficient)**	**r^2^ (model)**	** *p* ** **(model)**
1	Estradiol	3.526 (1.051)	0.478 (1.400; 5.653)	0.002	0.229	0.002
2	Estradiol	3.114 (1.021)	0.422 (1.044; 5.184)	0.004	0.315	0.001
	MEP	0.934 (0.433)	0.299 (0.057; 1.812)	0.038		
**Model: Birth weight in boys**		**B (SE)**	**β (95% CI)**	** *p* ** **(coefficient)**	**r^2^ (model)**	** *p* ** **(model)**
1	Estradiol	905.874 (369.396)	0.370 (158.071; 1653.678)	0.019	0.137	0.019
2	Estradiol	816.839 (354.654)	0.333 (98.243; 1535.436)	0.027	0.235	0.007
	MBzP	302.447 (138.392)	0.316 (22.039; 582.855)	0.035		
**Model: Gestational age at delivery in girls**		**B (SE)**	**β (95% CI)**	** *p* ** **(coefficient)**	**r^2^ (model)**	** *p* ** **(model)**
1	TSH	2.512 (0.872)	0.507 (0.711; 4.313)	0.008	0.257	0.008
2	TSH	2.850 (0.747)	0.575 (1.304; 4.395)	0.001	0.488	<0.001
	oxo-MiNP	1.868 (0.579)	0.486 (0.669; 3.066)	0.004		
**Model: Head circumference in girls**		**B (SE)**	**β (95% CI)**	** *p* ** **(coefficient)**	**r^2^ (model)**	** *p* ** **(model)**
1	FSH	−1.169 (0.517)	−0.434 (−2.240; −0.098)	0.034	0.189	0.034
2	FSH	−0.686 (0.349)	−0.255 (−1.411; 0.038)	0.062	0.670	<0.001
	cx-MiNP	2.533 (0.458)	0.716 (1.580; 3.486)	0.000		
3	FSH	−0.440 (0.323)	−0.164 (−1.114; 0.234)	0.188	0.753	<0.001
	cx-MiNP	2.218 (0.424)	0.627 (1.334; 3.102)	<0.001		
	OH-MnBP	0.935 (0.360)	0.322 (0.184; 1.687)	0.017		
4	FSH	−0.458 (0.294)	−0.170 (−1.073; 0.158)	0.136	0.806	<0.001
	cx-MiNP	2.600 (0.421)	0.735 (1.720; 3.481)	<0.001		
	OH-MnBP	1.128 (0.338)	0.388 (0.420; 1.837)	0.003		
	oxo-MEHP	−1.059 (0.466)	−0.273 (−2.034; −0.084)	0.035		

Legend: β—standardised coefficient, B—unstandardised coefficient, CI—confidence interval, FSH—follicle-stimulating hormone, FT4—thyroxin, MBzP—monobenzyl phthalate, MEP—monoethyl phthalate, OH-MiBP—mono(hydroxy-iso-butyl) phthalate, OH-MnBP—mono(hydroxy-n-butyl) phthalate, oxo-MEHP—mono(2-ethyl-5-oxohexyl) phthalate, oxo-MiNP—mono(oxo-methyl-octyl) phthalate, *p*—statistical significance value, r^2^—coefficient of determination, SE—standard error, SHBG—sex hormone-binding globulin, TSH—thyrotrophin.

## Data Availability

Not applicable.

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
