# Peer review of "Urinary Phthalate Biomarkers during Pregnancy, and Maternal Endocrine Parameters in Association with Anthropometric Parameters of Newborns"

_children, 2022, doi:10.3390/children9030413_

Round 1

Reviewer 1 Report

Dear Editor and authors,
I appreciated the opportunity to read and comment the manuscript titled “Urinary phthalate biomarkers during pregnancy, and maternal endocrine parameters in association with anthropometric parameters of newborn”.

This study aimed to investigate the mutual association between neonatal anthropometric parameters and maternal endocrine parameters during ≤15th week of gestation and 3rd trimester of pregnancy, concerning phthalate exposure of pregnant women.
This is the first study that indicate how prenatal exposure to phthalates may affect birth outcomes through interaction with the maternal endocrine system as a disruption and not only passing through the placental barrier, due to its lipophilic properties and intact directly the fetus, as studied before. The presented study is a part of the mother-Infant Study Court (PRENATAL) conducted in Nitra, Slovakia and the participation on the study was anonymous and voluntary; the population consisted of 72 mother-newborn pairs from October 2017 to December 2018.

The neonatal anthropometric parameters were gestational age at baby delivery, birth length and weight, head circumference and Apgar score; all the data were collected immediately after delivery. The body concentration of phthalates was collected by calculating urinary concentration of 17 phthalates metabolites. Furthermore, they determine Thyroid-stimulating hormone (TSH), thyroxine (T4), triiodothyronine (T3), follicle-stimulating hormone (FSH), luteinizing hormone (LH) estradiol (E2), testosterone (T), cortisol (F), and sex hormone-binding globulin (SHBG).

Using multivariate regression, they examined how phthalate exposure may affect the relationship between maternal endocrine parameters and birth outcomes in neonates: before they found associations when comparing the timing of phthalate exposure during pregnancy. During early stage and the 3rd trimester of pregnancy, they found a significant relationship between maternal endocrine parameters and gestational age at delivery, birth weight, and head circumference of newborns. Then they observed a significant relationship between maternal SHBG only during early pregnancy, and Apgar score 5 minutes after the delivery. At the end the statistical significance of hierarchical multivariate regression models increased after adding the concentrations of phthalate metabolites into models, demonstrating that there is an important dose-response effect.

However, the results of this study are very interesting. I think that the topic is interesting. References list appears up to date and appropriate. The manuscript is clear, the Spearman analysis for the study of the correlations is adequate and the results presented in ranks are clear and well descriptive for the phenomenon. The graphs and the tables are suitable for a quick and immediate comprehension and have a good descriptive legend suitable for a general reader without statistical and economical skills. It is very appreciable that after each acronym the meaning is always specified so that the reader does not have to spend time looking back for the meaning.
Anyway, I suggest some other revisions.

I suggest to better explain the characteristic of the cohorts such as age, parity, external expositions to environment, diet style. Above all it is important to describe if there are some comorbidities that could lead to endocrine system imbalance before and during pregnancy like hypothyroidism, hyperthyroidism, diabetes, hypertension, Chusing syndrome, Addison Syndrome, etc.

Iti is well known that birth outcomes in term of gestational age, birth weight and Apgar score could be highly influenced by these comorbidities and so their mention allows to avoid eventual biases.

I would also consider the area where the patients live and the pollution of water, air and soil. Furthermore, Slovakia could still have damage represented by radioactive contamination affecting, as already studied, the thyroid and other hormonal compartments.
I would also like a greater study and consequent discussion by the authors, they should analyze better the reason for the difference in effect on female and male fetus, why the results are so conflicting? Probably not only the fetal outcome should be investigated but also the imbalance in the mother and at what level the hormonal circuit is affected.

Author Response

Dear Reviewer,

We appreciate your time spent with reviewing our manuscript. We are grateful for your valuable comments and suggestions. All authors have carefully considered the comments. We have made revisions to the manuscript using the “Track Changes” function. Therefore, any changes are easily viewed. We hope the manuscript after careful revisions meet your standards. We provide our responses in the section below.

I suggest to better explain the characteristic of the cohorts such as age, parity, external expositions to environment, diet style. Above all it is important to describe if there are some comorbidities that could lead to endocrine system imbalance before and during pregnancy like hypothyroidism, hyperthyroidism, diabetes, hypertension, Cushing syndrome, Addison Syndrome, etc. It is well known that birth outcomes in term of gestational age, birth weight and Apgar score could be highly influenced by these comorbidities and so their mention allows to avoid eventual biases.

Response: Thank you for your suggestion. We have prepared separate table with data concerning mother cohort: age, BMI, number of previous pregnancies, education, living area, smoking, using hormonal therapy, using pharmaceuticals, disorders, total score in consumer practices (please, see Table 3). We have also added the separate sections in the results and discussion (chapters 3.4., and 4.2.).

I would also consider the area where the patients live and the pollution of water, air and soil. Furthermore, Slovakia could still have damage represented by radioactive contamination affecting, as already studied, the thyroid and other hormonal compartments.

Response: In order to minimize the effect of different environmental factors (water, soil, air) on phthalate exposure, we included to the study probands living only in the Nitra region. However, we have compared the exposure to phthalates in probands, whose living area is located in the urban areas with probands living in the rural areas (please, see Table 3). We have also added the separate sections in the results and discussion (please, see chapters 3.4., and 4.2.).

I would also like a greater study and consequent discussion by the authors, they should analyze better the reason for the difference in effect on female and male fetus, why the results are so conflicting? Probably not only the fetal outcome should be investigated but also the imbalance in the mother and at what level the hormonal circuit is affected.

Response: Thank you for your suggestion. We have compared the hormonal and phthalate metabolite concentrations in the mothers with male newborns and with mothers with female newborns (please, see chapters 3.5., 4.3., and 4.4.).

Reviewer 2 Report

Dear Authors,

In my opinion, the reviewed manuscript entitled ‘Urinary Phthalate Biomarkers during Pregnancy, and Maternal Endocrine Parameters in Association with Anthropometric Parameters of Newborn’ (ID: children-1602880)’ is interesting, well written and raises an important issue of exposure on endocrine disrupting chemicals (EDCs), which through dysregulation of the endocrine system, may affect the function of the whole organism, especially in fetal stage of life. Therefore, I believe that it is important to conduct research on exposure to EDCs. However, I have a few comments regarding the manuscript, which I listed below:

Introduction:

1) The introduction should contain more detailed information on the pathomechanism of EDCs and especially exposure on phthalate and its metabolites. Sentences in lines 53–59 does not fully explain the harmful effects of this specific EDCs. Moreover, the chemical structure of phthalates and their similarity to hormones has not been presented (please check: https://doi.org/10.3390/ijerph19042309 and https://doi.org/10.3389/fpubh.2020.00366).

Material and Methods

1) One of the strongest determinants of the course of pregnancy and the child's health is the age of the mother but there is no information about the average age or median and the range. In addition, the aging process will have an impact on the concentration of hormones that are tested in this manuscript.

2) Were there women in the study group who became pregnant spontaneously and thanks to the use of in vitro fertilization? The problem of exposure to EDCs in women undergoing fertility treatment due to the medications and treatments used is discussed. and it would be interesting to compare phthalate concentrations in body fluids of both groups of women (1 please check: 10.1016/j.scitotenv.2020.139834l; 10.1289/ehp.1509760; 10.1016/j.envres.2018.11.018)

3) What were the inclusion and exclusion criteria for the studies?

4) The criteria for the Apgar scale have not been provided.

5) In my opinion sentences from lines 102–106 should be moved to Acknowledge section of manuscript.

6) In section 2.5.: information about how the strength of the correlation was interpreted should be supplemented.

Results:

1) In table 2 please add information about size of groups (total, boys, girls).

2) In table 3: why the authors did not compare the differences between %≥LOQ between early pregnancy and 3th trimester? It is interesting to see if the concentration of phthalates significantly increases with the duration of pregnancy.

3) It is worth analyzing the correlation coefficients of age of women with the concentration of EDCs and other variables studied. This would additionally allow an analysis of the risk of bioaccumulation of phthalates in the body related to life expectancy.

Discussion:

Lines 385–386 ‘We hypothesize that phthalates as endocrine disruptors may affect birth outcomes through interaction with the maternal endocrine system’ without explaining how phthalates affect the endocrine system, this sentence is irrelevant.

Other:

1) Please consider adding a new section that will present the strengths and weaknesses of the research.

2) Minor corrections in the text are required, e.g. it should be mL instant ml (see table 4).

Author Response

Dear Reviewer,

We appreciate your time spent with reviewing our manuscript. We are grateful for your valuable comments and suggestions. All authors have carefully considered the comments. We have made revisions to the manuscript using the “Track Changes” function. Therefore, any changes are easily viewed. We hope the manuscript after careful revisions meet your standards. We provide our responses in the section below.

Introduction: The introduction should contain more detailed information on the pathomechanism of EDCs and especially exposure on phthalate and its metabolites. Sentences in lines 53–59 does not fully explain the harmful effects of this specific EDCs. Moreover, the chemical structure of phthalates and their similarity to hormones has not been presented (please check: https://doi.org/10.3390/ijerph19042309 and https://doi.org/10.3389/fpubh.2020.00366).

Response: Thank you for your recommendation. Part of the introduction have been rewritten as follows: “Therefore, our study aimed to determine this relationship since phthalates may act in the body by similar mechanisms as the natural hormones [12], such as competition for the binding site on transport proteins with hormones, alternation of synthesis, metabo-lism of steroid and non-steroid hormones, or alteration of a feedback loop within the hypothalamus, pituitary gland, and peripheral glands. The most studied effect of phthalates is the interaction with nuclear receptors. A possible reason is a fact that phthalates share a similar chemical structure with steroid hormones, and that is the benzene ring that binds to the receptor, acting like an agonist or antagonist (reviewed in [10], [13]). These interactions may result not only in complications and worsening of newborn's health but also can cause lifelong health problems.”

Material and Methods: One of the strongest determinants of the course of pregnancy and the child's health is the age of the mother but there is no information about the average age or median and the range. In addition, the aging process will have an impact on the concentration of hormones that are tested in this manuscript.

Response: Thank you for your suggestion. We have prepared separate table with data concerning mother cohort. Age of mothers is part of this table (please, see Table 3).

Material and Methods: Were there women in the study group who became pregnant spontaneously and thanks to the use of in vitro fertilization? The problem of exposure to EDCs in women undergoing fertility treatment due to the medications and treatments used is discussed. and it would be interesting to compare phthalate concentrations in body fluids of both groups of women (1 please check: 10.1016/j.scitotenv.2020.139834l; 10.1289/ehp.1509760; 10.1016/j.envres.2018.11.018).

Response: Thank you for your recommendation. Unfortunately, we did not ask probands about the use of in vitro fertilization in our questionnaire. However, we asked about use of pharmaceuticals and hormonal therapy in general (please, see Table 3).

Material and Methods: What were the inclusion and exclusion criteria for the studies?

Response: The exclusion criteria have been written as follows: “Exclusion criteria for the potential probands were age (≤17 years old), week of pregnancy during the early stage of pregnancy (≥16. week of pregnancy), risk pregnancy, hospitalization, severe disorder (e.g. cancer, stroke, heart attack), and living in other regions of Slovakia than Nitra.”

Material and Methods: The criteria for the Apgar scale have not been provided.

Response: The criteria for the Apgar score are provided in Table 1 (2.2 Birth data collection).

Material and Methods: In my opinion sentences from lines 102–106 should be moved to Acknowledge section of manuscript.

Response: We have changed the Acknowledgment section as follows: “We thank Michaela Foldesiova for her excellent technical assistance. We would like to thank the Institute for Prevention and Occupational Medicine of the German Social Accident Insurance (Bochum, Germany) for providing us with some of the analytical standards of phthalate metabolites as a generous gift.”

Material and Methods: In section 2.5.: information about how the strength of the correlation was interpreted should be supplemented.

Response: We have added some additional information about correlation coefficient and its strength as follows: “Correlations between maternal phthalate metabolites concentrations and birth outcomes were examined by Spearman correlation analysis. According to Evans [19], the strength of the correlation is determined by the height of the correlation coefficient (r): 0.00-0.19 is very weak, 0.20-0.39 is weak, 0.40-0.59 is moderate, 0.60-0.79 is strong, and 0.80-1.0 is very strong correlation.”

Results: In table 2 please add information about size of groups (total, boys, girls).

Response: We have added the suggested information.

Results: In table 3: why the authors did not compare the differences between %≥LOQ between early pregnancy and 3th trimester? It is interesting to see if the concentration of phthalates significantly increases with the duration of pregnancy.

Response: Thank you for your suggestion. We have compared the phthalate metabolite concentrations between early pregnancy and 3rd trimester of pregnancy (please, see Table 5). We have also added the sections in the results and discussion (chapters 3.2., and 4.1.).

Results: It is worth analyzing the correlation coefficients of age of women with the concentration of EDCs and other variables studied. This would additionally allow an analysis of the risk of bioaccumulation of phthalates in the body related to life expectancy.

Response: Thank you for your recommendation. We have analysed the age in relation to EDCs, endocrine parameters, and neonatal outcomes (please, see chapter 3.4., and 4.2.).

Discussion: Lines 385–386 ‘We hypothesize that phthalates as endocrine disruptors may affect birth outcomes through interaction with the maternal endocrine system’ without explaining how phthalates affect the endocrine system, this sentence is irrelevant.

Response: Thank you for your recommendation. Part of the discussion have been rewritten as follows: “We hypothesise that phthalates as endocrine disruptors may affect birth outcomes through interaction with the maternal endocrine system. It is known that phthalates in the body interface with the endocrine system at several levels, i.e., at the level of the hypothalamus, pituitary gland, peripheral organs, such as ovary, testis, thyroid, or ad-renal glands, transport proteins, and most significantly, phthalates can interact with nuclear receptors (reviewed in [10]). Therefore, we decided to analyse the mutual relationship between phthalate metabolites and endocrine parameters in our cohort using hierarchical multivariate regression.”

Other: Please consider adding a new section that will present the strengths and weaknesses of the research.

Response: We have added the new section (5. Strengths and limitations of study).

Other: Minor corrections in the text are required, e.g. it should be mL instant ml (see table 4).

Response: We have made the suggested changes in the whole manuscript.

Round 2

Reviewer 2 Report

Dear Authors,

I can consider that the manuscript has been properly corrected with regard to my comments. I have no more comments.